# Additional predictors of stroke and transient ischaemic attack in BEFAST positive patients in out-of-hours emergency primary care

**Florien S. van Royen**[1]*, **Geert-Jan Geersing**[1], **Daphne C. Erkelens**[1], **Mathé Delissen**[1], **Jorn V. F. Rutten**[1], **Dorien L. Zwart**[1], **Maarten van Smeden**[2,3], **Frans H. Rutten**[1], **Sander van Doorn**[1]

1 Department of General Practice & Nursing Science, Julius Center for Health Sciences and Primary Care, University Medical Center Utrecht, Utrecht University, Utrecht, the Netherlands, 2 Department of Epidemiology & Health Economics, Julius Center for Health Sciences and Primary Care, University Medical Center Utrecht, Utrecht University, Utrecht, the Netherlands, 3 Department of Data Science & Biostatistics, Julius Center for Health Sciences and Primary Care, University Medical Center Utrecht, Utrecht University, Utrecht, Netherlands

* f.s.vanroyen-5@umcutrecht.nl

**Data Availability Statement:** Data cannot be shared publicly because there are ethical

## Abstract

### Introduction

In patients suspected of stroke or transient ischemic attack (TIA), rapid triaging is imperative to improve clinical outcomes. For this purpose, balance-eye-face-arm-speech-time (BEFAST) items are used in out-of-hours primary care (OHS-PC). We explored the risk of stroke and TIA among BEFAST positive patients calling to the OHS-PC, and assessed whether additional predictors could improve risk stratification.

### Methods

This is a cross-sectional study of retrospectively gathered routine care data from telephone triage tape-recordings of patients calling the OHS-PC with neurological deficit symptoms, classified as BEFAST positive. Four models–with the predictors age, sex, a history of cardiovascular or cerebrovascular disease, and cardiovascular risk factors–were fitted using logistic regression to predict the outcome stroke or TIA. Likelihood ratio testing was used to select the best model, which was subsequently internally validated.

### Results

The risk of stroke or TIA diagnosis was 52% among 1,289 BEFAST positive patients, median age 72 years, 56% female sex. Of patients with the outcome stroke/TIA, 24% received a low urgency allocation, while 92% had signs or symptoms when calling. Only the addition of age and sex improved predicting stroke or TIA (internally validated c-statistic 0.72, 95%CI 0.69–0.75). The predicted risk of stroke or TIA remained below 20% in those aged below 40. Females aged 70 or over and males aged 55 or over, had a predicted risk above 50%.

restrictions being placed upon the data. The data contain potentially identifying and sensitive patient information. Moreover, a third party was involved in providing the telephone triage recordings from which the data were collected. This third party did not agree to make the data publicly available. The datasets generated during and/or analysed during the current study are available from the corresponding author on reasonable request. The data can also be requested by filling in the data request form from the University Medical Center Utrecht. The link is provided below. Research support staff from the division will review and consider the requests. https://preview.umcutrecht.nl/en/data-request-form-umc-utrecht.

**Funding:** This study was funded by an unrestricted grant from ZonMw (grant number 10060012210005). The Safety First Study was supported by the Department of General Practice of the University Medical Center Utrecht, the foundation 'Netherlands Triage Standard' and the foundation 'Stoffels-Hornstra'. The funders had no role in study design, data collection and analysis, decision to publish, or preparation of the manuscript.

**Competing interests:** The authors have declared that no competing interests exist.

**Abbreviations:** AUC, area under the curve; BEFAST, balance eye face arm speech time; CAD, cardiac artery disease; CI, confidence interval; CVD, cardiovascular disease; EHR, electronic healthcare record; FAST, face arm speech time; GDPR, General Data Protection Regulation; GP, general practitioner; ICPC, International Classification of Primary Care; IQR, interquartile range; MAR, missing at random; MNAR, missing not at random; NTS, Netherlands Triage Standard; OHS-PC, out-of-hours primary care; $R^2cs$, $R^2$ Cox-Snell; TIA, transient ischemic attack; U, urgency.

## Discussion

Urgency allocation appears to be suboptimal in BEFAST positive patients calling the OHS-PC. Risk stratification could be improved in this setting by adding age and sex.

## Introduction

Stroke and transient ischemic attack (TIA) are life-threatening medical emergencies that require rapid action to reduce morbidity and mortality. Nonetheless, early recognition remains difficult as stroke/TIA present themselves with many non-specific and often ambiguous symptoms. Moreover, there are important mimicking disorders such as migraine with aura, seizures, metabolic and toxic disorders, peripheral vestibular disease, Bell's palsy, collapse, and functional disorders [1–3]. In the Netherlands, many patients with such yet undefined symptoms will contact primary care first. During out-of-hours, this happens to be telephonically with the out-of-hours services in primary care (OHS-PCs) [4]. On the phone, patients are first assisted by a triage nurse supervised by a general practitioner (GP). The triage nurse assesses the urgency of the symptoms based on triage questions from the Netherlands Triage Standard (NTS), which is a semi-automatic decision support tool [5]. This allocation of urgency through triaging is a balancing act between safety and efficiency. Telephone triage needs to be safe enough to avoid missing cases of stroke/TIA (good sensitivity), while also being efficient enough by not unnecessarily assigning high urgency to low-risk patients that may overwhelm already strained healthcare utilisation during out-of-hours primary care (good specificity). Hence, triaging remains a difficult and challenging process [6].

The FAST items (face, arm, speech, time), later updated to BEFAST (adding balance and eye to FAST), were developed as diagnostic tools to create awareness about warning signs suggestive of stroke or TIA [7, 8]. The BEFAST items are mainly used and validated in the prehospital setting (ambulance dispatch centers and OHS-PC) and the emergency department of the hospital [9]. While these items are part of one of the 56 'entrance complaints' of the NTS, namely 'neurologic deficit', it is yet unclear what the implications are of using BEFAST items for the estimation of stroke/TIA risk through telephone triage in the primary care emergency setting [5].

This study describes the risk of stroke/TIA in callers to the OHS-PC with symptoms suggestive of neurological deficit and consequently classified as BEFAST positive. Furthermore, it aims to explore other simple clinical predictors to be used during telephone triage in BEFAST positive patients that can further aid in distinguishing between patients that have a higher risk of a diagnosis of stroke or TIA from those with a lower risk of having these diagnoses.

## Methods

### Study design and setting

This is a cross-sectional study using retrospective data from telephone triage tape-recordings of patients calling to the OHS-PC. In these facilities, triage nurses and GPs provide out-of-hours emergency primary care for all Dutch citizens. Data from nine OHS-PC locations in the central region of the Netherlands were used that provide out-of-hours care to 1.5 million inhabitants with 300,000 calls on average per year. This is a post-hoc analysis of the Safety First study and its design has been described in more detail elsewhere [10]. Data were accessed

from April 25 until November 17 2023. Where applicable, this study adhered to the TRIPOD checklist for prediction modelling studies [11].

## Study population

Between January 1 2014 and December 31 2017, a random sample of 2,500 recorded calls was selected for analysis. Patients with symptoms suggestive of stroke or TIA were identified based on a search of International Classification of Primary Care (ICPC) codes and/or keywords in the electronic healthcare records (EHR) of the OHS-PCs. Exclusion criteria for this study were age below 18, callers living outside OHS-PC area (final diagnosis not possible to retrieve), poor quality of recording and non-triage calls [10]. BEFAST classification was done after data collection. Those having at least one item scored as positive were considered 'BEFAST positive', and only BEFAST positive patients were included for further analyses. All ICPC codes and keywords used for inclusion and definitions of BEFAST items are provided in S1 Table.

## Data collection

Data were collected from the OHS-PC EHR and from telephone triage tape-recordings. Patient demographics and call characteristics were retrieved from the EHR and BEFAST items, medical history and final urgency allocation were collected from the phone tape-recordings. There are five urgency categories used by the OHS-PCs: U1 (ambulance dispatch within 15 min), U2 (GP consultation (home visit or at the OHS-PC) within one hour), U3 (GP consultation within three hours), U4 (GP consultation within 24 hours) and U5 (self-care telephone advice). All data were collected by trained researchers and medical students while blinded for the outcome.

## Outcome

The primary outcome of this study was a final diagnosis of stroke or TIA diagnosed by a neurologist or GP, the first with neuroimaging and the latter based on neurological deficit symptoms only. During triage and based upon urgency allocation, either an ambulance was sent, a home visit or consultation at the OHS-PC was offered, or the patient was advised to contact their own GP the next working day. Therefore, following clinical practice, the diagnostic work-up differed between patients. To ensure similar assessment of outcome for all patients, the final diagnosis was confirmed by the patient's own GP through discharge letters and medical record screening up to one month after the date of calling to the OHS-PC.

## Data analysis

Descriptive statistics were used for baseline characteristics. Categorical variables were summarised as numbers with percentages and continuous variables were summarised as means with standard deviations or medians with interquartile ranges. To identify additional predictors of stroke and TIA among BEFAST positive callers to the OHS-PC, four multivariable logistic regression models were fitted and compared by likelihood ratio tests (alpha of 0.05 for significance). A fixed modelling approach was used with predefined predictors based on literature and clinical experience. Model one consisted of the predictors age and sex. Model two additionally included a combined predictor for history of cardio- and/or cerebrovascular disease. Model three consisted of a combined predictor for cardiovascular risk factors including hypertension, diabetes and/or hypercholesterolaemia, in addition to age and sex. The fourth model included all predictors (age, sex, disease history of cardio- and cerebrovascular disease and cardiovascular risk factors). Age was handled as a continuous variable and a restricted cubic

spline with four knots on the percentiles 0.05, 0.35, 0.65 and 0.95 was applied to account for non-linearity. Additionally, an interaction term was added to age and sex [12]. Sex was handled as a categorical variable with two categories (biologically male and female) and all other predictors were also handled as categorical variables with two categories (e.g. history of cardio- and cerebrovascular disease being present or absent). To correct for optimism in the model's estimates, the best model was internally validated using bootstrapping with 100 repetitions after which the area under the curve (AUC/c-statistic), $R^2$ and slope were calculated. Statistical analyses were performed in R version 4.2.2 with R base, rms, mice and pROC packages [13–16].

## Sample size considerations

For sample size calculation, the method by Riley et al. for logistic regression modelling was used [17]. For prediction model development, 1,289 BEFAST positive patients were available with an outcome rate of 0.52 for the combined outcome of stroke and TIA. A c-statistic of 0.70 was chosen for sample size calculation which was based on the c-statistic reported in a similar study validating a TIA recognition tool in primary care [18]. The sample size of 1,289 patients was calculated to be large enough to include up to a maximum of 18 candidate predictors. This was sufficient to include the prespecified predictors as described above, restricted cubic spline for age with four knots and an interaction term for age and sex.

## Missing data

Missing data for BEFAST items were not imputed because of the complexity of data structure, i.e. the presence of BEFAST items is dependent on ordered questions and answers on previous triage questions. Hence, we assumed missing data on BEFAST to follow a MNAR (missing not at random) pattern. It is widely acknowledged that in such circumstances it is preferred to refrain from imputation of these items [19]. Missing data for candidate predictors were assumed to be MAR (missing at random) and were imputed using multiple imputation by chained equation methods included in the 'mice' package in R. A random forests method was used, and 100 datasets were generated with 20 iterations [13, 20]. The percentage of missing data per predictor is shown in S2 Table.

## Ethical approval

This study was conducted in accordance with Dutch law, the European Union General Data Protection Regulation (GDPR) and the principles of the Declaration of Helsinki. It is part of the larger Safety First study (National Trial Register identification number: NTR7331) [10]. The Medical Ethics Review Committee Utrecht, the Netherlands, reviewed the study and formal approval was waived as minimal patient participation was required. During data collection from telephone recordings, data were pseudonymised for further analyses conform the GDPR.

## Results

### Population

From the random sample of 2,500 recorded calls that were selected based on the inclusion criteria, 1,381 could be used for final analyses. The other 1,119 calls were excluded based on exclusion criteria or because the outcome could not be retrieved due to nonresponse or refusal of the enlisted GP. Details on patient flow through the study are depicted in S1 Fig. After data collection, 1,289 patients were classified as BEFAST positive. There were 92 patients for whom

BEFAST could not be determined because of missing data, and these patients were excluded from further analyses.

## Patient characteristics

Characteristics of all BEFAST positive patients are shown in Table 1. Median age was 72 years, 56% were female and 92% still had symptoms during the call to the OHS-PC. In 52% of stroke or TIA was diagnosed; 17% stroke and 35% TIA or minor stroke. These patients were generally older (median 79 years versus median 64 years), more often had a history of cardiovascular disease and had more cardiovascular risk factors (hypertension, hypercholesterolaemia and diabetes) than those without stroke or TIA. 24% of BEFAST positive patients with stroke or TIA received a low urgency allocation during triage (U3, U4 or U5), these patients had similar demographic characteristics (median age 80 versus 79 years and 58% versus 55% female sex), more often had a personal history of TIA (37% versus 32%) and less often had a personal history of stroke (23% versus 28%) than patients that were correctly allocated to high urgency (U1 and U2). A neurologist diagnosed stroke or TIA in 83% of the cases based on clinical symptoms plus neuroimaging, and 17% (mainly older patients) was diagnosed by GPs based on clinical symptoms.

## Predictors of stroke and TIA in BEFAST positive patients

All 1,289 BEFAST positive patients were used to develop the four prespecified models. When compared by likelihood ratio test, in our dataset, model 2, 3 and 4 (including the added predictors history of cardio- and cerebrovascular disease and cardiovascular risk factors) did not prove to perform significantly better than model 1 only including age and sex, as shown in

**Table 1. Patient characteristics of all BEFAST positive patients.**

| Patient characteristic | BEFAST positive patients (n = 1,289) | Outcome no stroke/TIA (n = 617, 48%) | Outcome stroke/TIA (n = 672, 52%) |
|---|---|---|---|
| **Age in years (IQR)** | 72 (58–86) | 64 (47–78) | 79 (68–86) |
| **Female sex** | 728 (56%) | 355 (58%) | 373 (56%) |
| **History of TIA** | 182 (28%, n = 658) | 62 (21%, n = 297) | 120 (33%, n = 361) |
| **History of stroke** | 175 (27%, n = 658) | 78 (26%, n = 297) | 97 (27%, n = 361) |
| **History of CVD** | 685 (77%, n = 889) | 279 (68%, n = 409) | 409 (85%, n = 483) |
| **History of CAD** | 55 (18%, n = 310) | 26 (15%, n = 174) | 29 (21%, n = 136) |
| **History of arrythmia** | 60 (26%, n = 299) | 22 (13%, n = 167) | 38 (29%, n = 132) |
| **Heart failure** | 24 (9%, n = 269) | 8 (5%, n = 155) | 16 (14%, n = 114) |
| **Hypertension** | 212 (49%, n = 430) | 86 (38%, n = 224) | 126 (61%, n = 206) |
| **Hypercholesterolaemia** | 167 (42%, n = 400) | 62 (31%, n = 202) | 105 (53%, n = 198) |
| **Diabetes** | 149 (35%, n = 424) | 62 (28%, n = 226) | 87 (44%, n = 198) |
| **Acute onset of symptoms** | 64 (30%, n = 212) | 32 (29%, n = 111) | 32 (32%, n = 101) |
| **Symptoms still present at time of calling** | 1185 (92%) | 573 (93%) | 612 (91%) |
| **U1 urgency** | 316 (25%) | 126 (20%) | 190 (28%) |
| **U2 urgency** | 591 (46%) | 268 (43%) | 323 (48%) |
| **Low urgency (U3, U4 or U5)** | 385 (30%) | 223 (36%) | 159 (24%) |
| **Referred to neurologist** | 878 (69%, n = 1275) | 323 (53%, n = 609) | 555 (83%, n = 666) |
| **Final diagnosis TIA/minor stroke** | | | 455 (68%) |
| **Final diagnosis major stroke** | | | 217 (32%) |

BEFAST = balance eye face arm speech time; CAD = cardiac artery disease; CVD = cardiovascular disease; IQR = interquartile range; TIA = transient ischaemic attack; U = urgency

**Table 2. Comparison of the four models.**

|  | Model 1 versus model 2 | Model 1 versus model 3 | Model 1 versus model 4 |
|---|---|---|---|
| **ΔAUC** | 0.001 | 0.001 | 0.001 |
| **Likelihood ratio test** | 0.933, df = 1, p = 0.334 | 0.448, df = 1, p = 0.503 | 0.413, df = 2, p = 0.662 |

ΔAUC and likelihood ratio test comparing model 2 (including the predictor history of cardio- and cerebrovascular disease), model 3 (including the predictor cardiovascular risk factors) and model 4 (including both the predictors history of cardio- and cerebrovascular disease and cardiovascular risk factors) with model 1 (including only the predictors age and sex). ΔAUC is calculated by taking the differences between the two models unadjusted c-statistics.

Table 2. The regression coefficients with confidence intervals, the apparent performance and internal validation performance of model 1 are shown in Table 3. Regression coefficients with confidence intervals and apparent performance of model 2, model 3 and model 4 are provided in S3 Table. The apparent c-statistic of model 1 was 0.73 (95% CI 0.70–0.75) and the internally validated c-statistic was 0.72 (95%CI 0.69–0.75). In Fig 1, the predicted risks of stroke and TIA predicted by model 1 are plotted for age, and male and female sex. Overall, predicted risk increased with age and was higher for male patients at mid-life than for female patients at mid-life. For instance, until the age of 40, the predicted risk for both female and male patients remained below 20%. A risk of >50% was reached for female patients from the age of about 70, while for male patients this was reached from the age of about 55.

## Discussion

This study showed that patients calling to the OHS-PC with symptoms suggestive of stroke or TIA, and at least one item of BEFAST positive, had a high risk (52%) of having a final diagnosis of stroke or TIA. Of these stroke/TIA patients, 76% received a high urgency allocation. Only sex and age could improve triaging while a history of cardio- and cerebrovascular disease, or

**Table 3. Model development and internal validation of model 1 using multivariable logistic regression.**

| Predictor | Regression coefficient | 95% CI |
|---|---|---|
| **Intercept** | -5.993 | -8.526; -3.459 |
| **Age** | 0.115 | 0.063; 0.167 |
| **Age'** | -0.111 | -0.185; -0.036 |
| **Age"** | 0.690 | 0.128; 1.252 |
| **Female sex** | 2.255 | -0.823; 5.333 |
| **Interaction sex and age** | -0.063 | -0.128; 0.001 |
| **Interaction sex and age'** | 0.114 | 0.020; 0.208 |
| **Interaction sex and age"** | -0.634 | -1.356; 0.078 |
|  | **Performance measures** |  |
| **Apparent c-statistic** | 0.73 | 0.70; 0.75 |
| **$R^2$cs** | 0.22 |  |
| **Internal validation c-statistic** | 0.72 | 0.69; 0.75 |
| **Internal validation $R^2$** | 0.21 |  |
| **Internal validation slope** | 0.96 |  |

Regression coefficients with 95% confidence intervals, c-statistic, and internal validation performance measures of the best model (model 1) including the predictors age, sex and the interaction between age and sex. Age was divided into three subgroups (shown as age, age' and age") using restricted cubic spline function to account for non-linearity. CI = confidence interval; $R^2$cs = $R^2$ Cox-Snell.

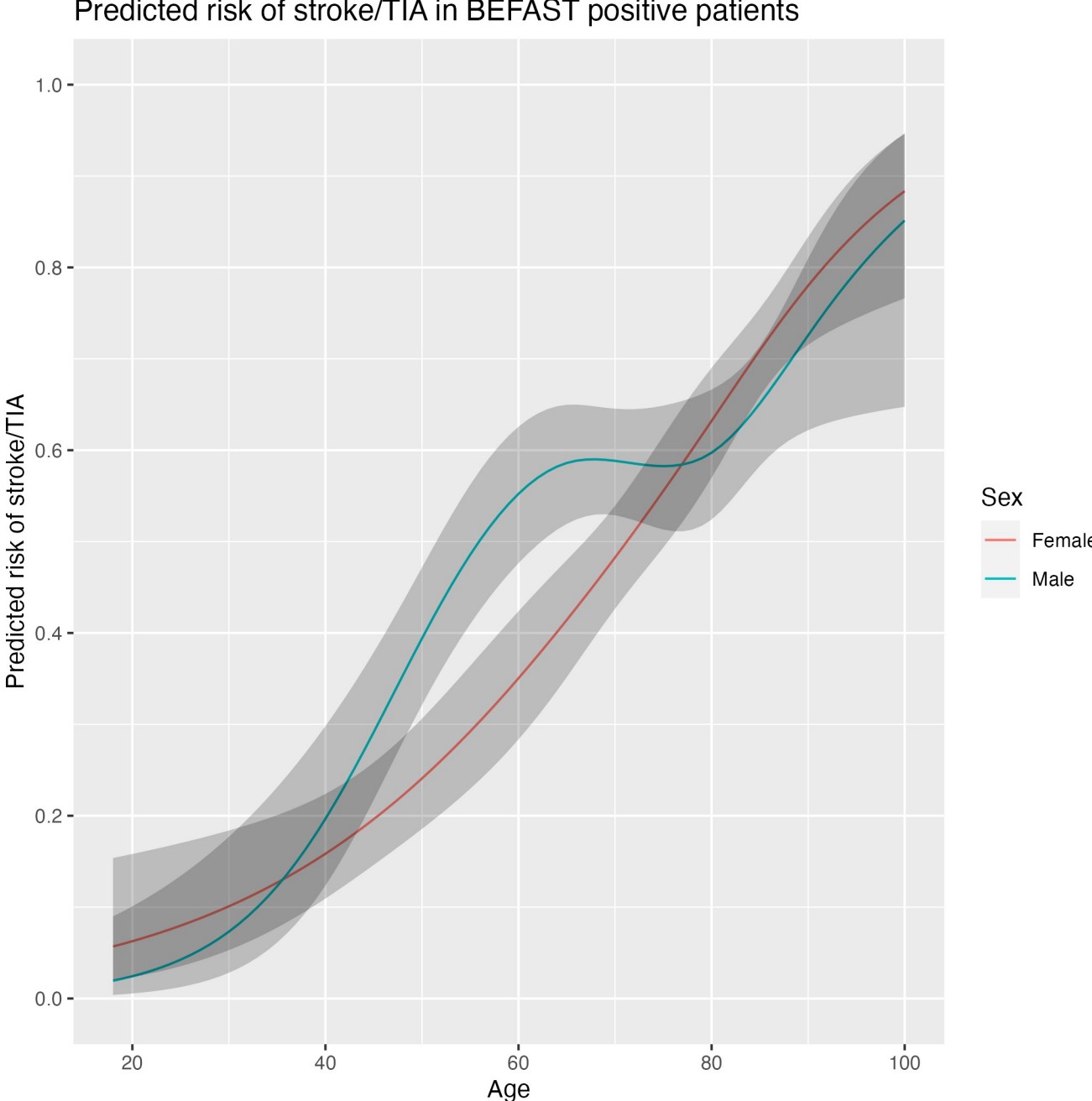

**Fig 1. Predicted risk of stroke and TIA.** Plot of predicted risk of stroke and TIA in BEFAST positive patients shown for men and women at different ages. Confidence intervals are shown in grey.

cardiovascular risk factors (hypertension, diabetes or hypercholesterolaemia) could not further aid in predicting stroke/TIA in this high-risk patient group.

## Comparison with existing literature

In the prehospital setting, such as the OHS-PC, where patients with possible stroke/TIA often present early-on in the disease course, it is pivotal to differentiate life-threatening conditions, such as stroke/TIA, from more benign conditions and mimics such as migraine with aura.

During initial telephone triage, an accurate definitive diagnose may not yet be necessary, but accurate urgency allocation of patients at (high) risk of stroke or TIA is crucial. Therefore, any triaging tool needs a sufficiently high sensitivity (i.e. not missing any cases) while a high specificity is needed for efficiency. A Cochrane review from 2019 assessed the diagnostic accuracy of the FAST items and identified three studies reporting a sensitivity in the prehospital setting ranging from 0.64 to 0.97, however, with a broad range in specificity from 0.13 to 0.75 [9]. The addition of balance disturbance (B) and eye problems (E) to FAST is supposed to prevent missing stroke or TIA of the posterior cerebral circulation and thus increase sensitivity, however adding items to FAST will certainly further decrease specificity and thus efficiency [8]. By design, we selected patients that were all classified as BEFAST positive in our study sample, therefore, it was not feasible to calculate sensitivity and specificity for the BEFAST triaging tool.

Currently, there are many stroke and TIA prediction models and diagnostic tools available, mainly to be used at the emergency department or to detect large vessel occlusions to direct intervention [9, 21, 22]. Almost all these tools use similar items to BEFAST (i.e. signs and symptoms) to assess the probability of stroke or TIA, lacking possible additional predictors such as age, sex, cardio-cerebrovascular history and cardiovascular risk factors. In our study, sex and age provided additive predictive information in a BEFAST positive population while a history of cardio- or cerebrovascular disease, or cardiovascular risk factors (hypertension, diabetes or hypercholesterolaemia) did not add predictive information beyond signs and symptoms of BEFAST. Apparently, in a population already selected for their signs and symptoms and therefore with a high a priori risk of a stroke/TIA diagnosis, other clinical factors besides adding age and sex, will not further increase the ability to distinguish between higher and lower risk. We also assessed 'someone else calling the OHS-PC for the patient' as a predictor for stroke or TIA, but this variable had no added value beyond the predictors age and sex (data not shown).

## Strengths and limitations

The major strength of this research is the use of routine care data, reflecting real-world clinical practice. Furthermore, only readily available predictors were analysed, preventing the increase in workload of triage nurses and GPs. Formal external validation of the final prediction model should be conducted to confirm predictive performance over time in the Dutch OHS-PC setting or to prove its predictive performance in other settings.

Two limitations to this work must be discussed. First, the dataset had missing values, a common finding when using routine care data. Although imputations were carefully executed and only under the MAR assumptions, introduction of some bias cannot be fully ruled out. Importantly, BEFAST is only considered negative if none of the symptoms are present, which was in none of our patients (partly due to missing data). As a result, we could not calculate the sensitivity, specificity, and predictive values of BE-FAST. Therefore, our results should be interpreted with some caution, and studies repeating our analyses in new data are necessary to confirm our findings. Second, the outcome of stroke and TIA was not assessed similarly for all patients, since only the patients that were still considered to be at risk of stroke or TIA after GP consultation were referred to the neurologist for further diagnostic imaging (differential verification) [11]. However, it is unlikely that many stroke or TIA cases were missed as a consequence of this selective referral. To make it even more unlikely to miss cases of stroke or TIA and to assess the outcome for all patients in a similar way, the outcome was assessed through contacting the enlisted GPs up to one month after visiting the OHS-PC, which also encompasses, for instance, delayed discharge letters.

## Clinical implications

This study adds to a better understanding of the distribution of risks among BEFAST positive patients (i.e. the patients suspected of stroke or TIA) at the triage stage. While 92% of BEFAST positive patients still had symptoms at the moment of calling and 52% eventually received a diagnosis of stroke or TIA, only 76% of patients with the outcome stroke/TIA received a high urgency allocation. Incorporating age and sex of the patient in the triage process may improve urgency allocation, e.g. upscaling the urgency for elderly patients (both sexes) and for middle-aged male patients. External validation studies (temporal as well as geographical) and implementation studies are needed to evaluate the generalizability and clinical impact of these findings. Moreover, additional strategies to improve urgency allocation in patients suspected of stroke or TIA may be considered. For instance, it has been shown that safety of telephone triage improved when triage nurses overruled the decision support system (NTS), and urgency allocation was adjusted, either or not after consultation of a GP [6]. Such strategies may be the key to sustainable, safe and efficient telephone triage.

## Conclusion

For BEFAST positive patients calling to the OHS-PC, the risk of stroke or TIA was above 50%. Despite this high risk, one in four patients received a low urgency allocation. The addition of age and sex can improve risk stratification with higher risks observed in older and male patients.

## Supporting information

**S1 Fig. Patient flow through the study.**
(TIF)

**S1 Table. ICPC codes, keywords and definition of BEFAST items used in this study.**
ICPC = International Classification of Primary Care.
(DOCX)

**S2 Table. Percentage of missing data per predictor in BEFAST positive patients.**
(DOCX)

**S3 Table. Model development and apparent performance of model 2, model 3 and model 4 using multivariable logistic regression.** Regression coefficients with 95% confidence intervals, c-statistic, and apparent performance measures of model 2, model 3 and model 4. Age was divided into three subgroups (shown as age, age' and age") using restricted cubic spline function to account for non-linearity. *The predictor history of cardio- and cerebrovascular disease is a combination of history of stroke, history of TIA and history of cardiovascular disease. **The predictor cardiovascular risk factors is a combination of diabetes, hypercholesterolaemia and hypertension. CI = confidence interval.
(DOCX)

## Acknowledgments

The authors thank the OHS-PC foundation 'Primair Huisartsenposten' and all medical students that contributed to data collection.

## Author Contributions

**Conceptualization:** Florien S. van Royen, Geert-Jan Geersing, Maarten van Smeden, Frans H. Rutten, Sander van Doorn.

**Data curation:** Daphne C. Erkelens, Mathé Delissen, Jorn V. F. Rutten, Dorien L. Zwart.

**Formal analysis:** Florien S. van Royen.

**Funding acquisition:** Dorien L. Zwart, Frans H. Rutten, Sander van Doorn.

**Investigation:** Florien S. van Royen.

**Methodology:** Florien S. van Royen, Geert-Jan Geersing, Maarten van Smeden, Sander van Doorn.

**Resources:** Daphne C. Erkelens, Mathé Delissen, Jorn V. F. Rutten, Dorien L. Zwart.

**Supervision:** Geert-Jan Geersing, Frans H. Rutten, Sander van Doorn.

**Visualization:** Florien S. van Royen.

**Writing – original draft:** Florien S. van Royen.

**Writing – review & editing:** Geert-Jan Geersing, Daphne C. Erkelens, Mathé Delissen, Jorn V. F. Rutten, Dorien L. Zwart, Maarten van Smeden, Frans H. Rutten, Sander van Doorn.

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
