## [Decision Letter · Decision Letter 0]

8 May 2024

PONE-D-23-43803Additional predictors of stroke and transient ischaemic attack in BEFAST positive patients in out-of-hours emergency primary carePLOS ONE

Dear Dr. van Royen,

Thank you for submitting your manuscript to PLOS ONE. After careful consideration, we feel that it has merit but does not fully meet PLOS ONE’s publication criteria as it currently stands. Therefore, we invite you to submit a revised version of the manuscript that addresses the points raised during the review process. Most of the points raised by the reviewers are addressable.

We look forward to receiving your revised manuscript.

Kind regards,

Shashank Shekhar, MD

Academic Editor

PLOS ONE

Journal Requirements:

2. Thank you for stating the following financial disclosure: "This study was funded by an unrestricted grant from ZonMw (grant number 10060012210005). The Safety First Study was supported by the Department of General Practice of the University Medical Center Utrecht, the foundation ‘Netherlands Triage Standard’ and the foundation ‘Stoffels-Hornstra’."

3. In the online submission form, you indicated that the datasets generated during and/or analysed during the current study are available from the corresponding author on reasonable request.

Reviewers' comments:

Reviewer's Responses to Questions

**Comments to the Author**

1. Is the manuscript technically sound, and do the data support the conclusions?

Reviewer #1: Yes

Reviewer #2: Partly

2. Has the statistical analysis been performed appropriately and rigorously? 

Reviewer #1: Yes

Reviewer #2: Yes

3. Have the authors made all data underlying the findings in their manuscript fully available?

Reviewer #1: No

Reviewer #2: Yes

4. Is the manuscript presented in an intelligible fashion and written in standard English?

Reviewer #1: Yes

Reviewer #2: Yes

5. Review Comments to the Author

Reviewer #1: In patients suspected of stroke or transient ischemic attack (TIA), rapid triaging is imperative to improve clinical outcomes. For this purpose, balance-eye-face- arm-speech-time (BEFAST) items are used in out-of-hours primary care (OHS-PC). In this study, the authors investigated the risk of stroke and TIA among BEFAST positive patients calling to the OHS-PC, and assessed whether additional predictors could improve risk stratification. They report that the risk stratification could be improved in this setting by adding age and sex.

Reviewer #2: This is an interesting cross-sectional study conducted in Netherlands evaluating the addition of age and sex as predictors of stroke/TIA in BE-FAST positive patients in a pre-hospital setting. Although the design of the study, methodology used, interpretation of results and conclusions drawn has been impressive; the following concerns/issues remain unaddressed:

1. Introduction, Line 78 - "Moreover, there are important mimicking disorders such as migraine with aura.(1–3)" Stroke mimics are not limited to migraine with aura only. Physicians also need to be consider other common stroke mimics like syncope, seizures, hypoglycemia, etc., as important differentials. Consider adding these things when talking about stroke mimics in this statement.

2. Study design, Line 105 - "cross-sectional study". As per the methodology discussed in the manuscript, this would qualify as a retrospective cross-sectional study specifically. Please correct.

3. Outcome, Line 134 - "The primary outcome of this study was a final diagnosis of stroke or TIA diagnosed by a neurologist or GP." Can the authors provide more details on how the diagnosis of stroke/TIA was established? Was it based on clinical symptoms, neuroimaging or both? Please modify as appropriate.

4. Results, line 207 - "35% TIA or minor stroke." How many of these were moderate or high risk TIA? Please include in the results.

5. Discussion, Line 281 - " By design, we selected patients that were all classified as BEFAST positive in our study sample, therefore, it was not feasible to calculate sensitivity and specificity for the BEFAST triaging tool." It would have been more useful to see the sensitivity and specificity of BE-FAST in this patient population and to draw a meaningful conclusion on the performance of BE-FAST scale as compared to some of the other numbers in prior literature. The authors had access to all the records of stroke/TIA diagnosis to begin with. Any particular reason as to why only BE-FAST positive patients were selected for this study which probably limited the ability to calculate sensitivity, specificity, PPV and NPV in this patient population? Please explain in the discussion or limitations subheading.

6. Conclusion - " The addition of age and sex can improve risk stratification with higher risks observed in older and male patients." The median age noted was almost similar between the low urgency and high urgency group (80 vs. 79 years). Yet, age was noted to be a significant predictor of stroke/TIA in BE-FAST positive patients. Please explain?

7. The authors did not discuss if addition of age and sex to BE-FAST based on the results of this single center study would be applicable to patients from another geographic location - South East Asia (higher prevalence of elderly population in general along with vascular risk factors, African American and hispanic population (earlier age of onset of stroke and higher prevalence of atherosclerosis)? Please discuss under implications of this study.

6. PLOS authors have the option to publish the peer review history of their article (what does this mean?). If published, this will include your full peer review and any attached files.

Reviewer #1: **Yes: **Aurel Popa-Wagner

Reviewer #2: No

---

## [Author Response · Author response to Decision Letter 0]

30 Jul 2024

Shashank Shekhar, MD

Academic Editor

PLOS ONE

Dear Editor, 

We are very grateful for the opportunity given to resubmit our manuscript entitled ‘Additional predictors of stroke and transient ischaemic attack in BEFAST positive patients in out-of-hours emergency primary care’. We thank the editor and reviewers for the useful suggestions which we incorporated in our revised manuscript. Please find below the point-by-point answers to the questions and the changes made in the revised manuscript, which are also marked as ‘track changes’.

We are looking forward to your response to our revised manuscript. 

On behalf of the co-authors,

Kind regards, 

Florien S. van Royen, MSc, GP trainee

Dept. General Practice & Nursing Science, Julius Center for Health Sciences and Primary Care, UMC Utrecht, Utrecht University, Utrecht, the Netherlands

Journal Requirements

The manuscript has now been formatted according to the style requirements as requested by PLOS One. 

2. Thank you for stating the following financial disclosure: "This study was funded by an unrestricted grant from ZonMw (grant number 10060012210005). The Safety First Study was supported by the Department of General Practice of the University Medical Center Utrecht, the foundation ‘Netherlands Triage Standard’ and the foundation ‘Stoffels-Hornstra’."

The proposed statement "The funders had no role in study design, data collection and analysis, decision to publish, or preparation of the manuscript" is correct and has been included in our cover letter. 

3. In the online submission form, you indicated that the datasets generated during and/or analysed during the current study are available from the corresponding author on reasonable request.

We agree with PLOS that full data transparency is preferred to support scientific findings. However, the data agreement contract with the third-party providing access to the pseudonymized data supporting this study did not include any conditions for publishing the data open access. Our local Medical Ethics Review Committee also did not ask us to provide to publish our data on an open-access platform, nor did they provide conditions for this. Therefore, at present, it is not possible to ask the parties involved for permission in hindsight. Because our data are highly privacy-sensitive, including backed-up telephone conversations with patients, and because an opt-out procedure was used for data collection, it is currently not possible to permit the publication of our dataset. We hope that the editor understands our reasons for not being able to make the dataset publicly available. We are happy to further discuss this with the editor and we are open to suggestions to increase transparency regarding the data underlying our findings. 

Review Comments to the Author

Reviewer #1: In patients suspected of stroke or transient ischemic attack (TIA), rapid triaging is imperative to improve clinical outcomes. For this purpose, balance-eye-face- arm-speech-time (BEFAST) items are used in out-of-hours primary care (OHS-PC). In this study, the authors investigated the risk of stroke and TIA among BEFAST positive patients calling to the OHS-PC, and assessed whether additional predictors could improve risk stratification. They report that the risk stratification could be improved in this setting by adding age and sex.

We thank the reviewer for appraisal of the main message of our study. 

Reviewer #2: This is an interesting cross-sectional study conducted in Netherlands evaluating the addition of age and sex as predictors of stroke/TIA in BE-FAST positive patients in a pre-hospital setting. Although the design of the study, methodology used, interpretation of results and conclusions drawn has been impressive; the following concerns/issues remain unaddressed:

1.Introduction, Line 78 - "Moreover, there are important mimicking disorders such as migraine with aura.(1–3)" Stroke mimics are not limited to migraine with aura only. Physicians also need to be consider other common stroke mimics like syncope, seizures, hypoglycaemia, etc., as important differentials. Consider adding these things when talking about stroke mimics in this statement.

Answer: 

Indeed, there are many stroke mimics. For clarifying reasons, we added more examples of common stroke mimics to this statement, see below. 

Changes to the manuscript:

This statement now reads: ‘Moreover, there are important mimicking disorders such as migraine with aura, seizures, metabolic and toxic disorders, peripheral vestibular disease, Bell’s palsy, collapse, and functional disorders.(1–3)’

2. Study design, Line 105 - "cross-sectional study". As per the methodology discussed in the manuscript, this would qualify as a retrospective cross-sectional study specifically. Please correct.

Answer: 

Thank you for noticing that the manuscript does not mention the retrospective nature of the data collection. 

Changes to the manuscript:

- We have changed the wording in the methods to: ‘This is a cross-sectional study using retrospective data from telephone triage tape-recordings of patients calling to the OHS-PC.’

- We have changed the wording in the abstract to: ‘This is a cross-sectional study of retrospectively gathered routine care data from telephone triage tape-recordings of patients calling the OHS-PC with neurological deficit symptoms, classified as BEFAST positive.’

3. Outcome, Line 134 - "The primary outcome of this study was a final diagnosis of stroke or TIA diagnosed by a neurologist or GP." Can the authors provide more details on how the diagnosis of stroke/TIA was established? Was it based on clinical symptoms, neuroimaging or both? Please modify as appropriate.

In the 83% referred to the neurologist, the diagnosis was based on clinical symptoms plus neuroimaging, thus following the full diagnostic assessment of the treating neurologist. The diagnosis in the other 17% (mainly older patients), was based on the GP’s assessment of clinical symptoms. 

Changes to the manuscript

- We changed this sentence in the Methods to: ‘The primary outcome of this study was a final diagnosis of stroke or TIA diagnosed by a neurologist or GP, the first with neuroimaging and the latter based on neurological deficit symptoms only.’ 

- We changed the last sentence of the paragraph describing the patient characteristics in the Results to: ‘A neurologist diagnosed stroke or TIA in 83% of the cases based on clinical symptoms plus neuroimaging, and 17% (mainly older patients) was diagnosed by GPs based on clinical symptoms.’

4. Results, line 207 - "35% TIA or minor stroke." How many of these were moderate or high risk TIA? Please include in the results.

Answer:

Unfortunately, this information was not available in the routine care data. TIA and minor stroke were considered and presented as a single variable during data collection and writing. 

5. Discussion, Line 281 - " By design, we selected patients that were all classified as BEFAST positive in our study sample, therefore, it was not feasible to calculate sensitivity and specificity for the BEFAST triaging tool." It would have been more useful to see the sensitivity and specificity of BE-FAST in this patient population and to draw a meaningful conclusion on the performance of BE-FAST scale as compared to some of the other numbers in prior literature. The authors had access to all the records of stroke/TIA diagnosis to begin with. Any particular reason as to why only BE-FAST positive patients were selected for this study which probably limited the ability to calculate sensitivity, specificity, PPV and NPV in this patient population? Please explain in the discussion or limitations subheading.

Answer:

This is a valid question and in the initial design of our study, we did plan to evaluate the sensitivity, specificity, PPV and NPV of the BEFAST tool for telephone triage in out-of-hours primary care (OHS-PC). However, none of the included patients suspected of stroke or TIA (based on International Classification of Primary Care (ICPC) codes and/or keywords in the electronic healthcare records (EHR) of the OHS-PCs) in our random sample of callers were classified as BEFAST negative. All had at least one BEFAST positive variable (balance disturbance, eye symptoms, face drooping, arm symptoms, speech problems). 

To be more precise (see also the population section of our results), the initial dataset contained 1,381 patients suspected of stroke or TIA based on our inclusion criteria. 1,289 (93%) of these patients were considered BEFAST positive based on the presence of one or more symptoms. 92 (7%) of these patients had at least one missing variable on BEFAST and all other symptoms scored absent. However, because all variables should be negative to classify an individual as BEFAST negative and because we refrained from imputation of BEFAST variables due to data complexity (as explained in the methods section), eventually no included patients could be classified as BEFAST negative. Likely, this is also a consequence of the setting where we recruited patients, namely patients calling the OHS-PC service for complaints that lead to a suspicion of TIA/stroke, in fact because they experienced ‘BEFAST symptoms’. Put simply, patients without ‘BEFAST symptoms’ simply did not call the OHS-PC service and thus were not included in our dataset. 

For this reason, we did not have a ‘test (in this case BEFAST) negative’ group to calculate sensitivity, specificity, PPV and NPV for BEFAST in this setting. In our paper, we, therefore, focus on exploring the risk of stroke or TIA in BEFAST-positive callers as this is the patient group that is calling OHS-PC for subsequent advice and/or management. Subsequently, our aim was to improve urgency allocation and efficiency of telephone triage in the out-of-hours primary care setting by exploring additional variables that could be further used to differentiate between high- and low-risk patients for having TIA/stroke. We found that the risk of stroke or TIA diagnosis was high (52%) among 1,289 BEFAST positive patients and that only the addition of age and sex improved predicting stroke or TIA in this group. 

Changes to the manuscript:

We added the following sentence about sensitivity, specificity, NPV and PPV to the limitations section: ‘Importantly, BEFAST is only considered negative if none of the symptoms are present, which was in none of our patients (partly due to missing data). As a result, we could not calculate the sensitivity, specificity, and predictive values of BE-FAST.’

6. Conclusion - " The addition of age and sex can improve risk stratification with higher risks observed in older and male patients." The median age noted was almost similar between the low urgency and high urgency group (80 vs. 79 years). Yet, age was noted to be a significant predictor of stroke/TIA in BE-FAST positive patients. Please explain?

Answer: 

The reviewer refers to the median ages of patients of the specific groups correctly classified as ‘high urgency’ or incorrectly as ‘low urgency’ in the group of patients with the outcome present (diagnosis of stroke or TIA, n=672). These groups do not differ in median age. In the main analysis, the median age of patients with and without the outcome (n=1,289) is used as a predictor, which differed substantially (median age 79 years in those with TIA/stroke versus 64 years in those without TIA/stroke, respectively).

It should also be noted that the numbers provided in the baseline table reflect ‘univariable’ associations between variables and the outcome stroke or TIA. These associations may differ from the associations in the context of other variables as is investigated through multivariable analysis including interaction terms and spline functions. 

7. The authors did not discuss if addition of age and sex to BE-FAST based on the results of this single center study would be applicable to patients from another geographic location - South East Asia (higher prevalence of elderly population in general along with vascular risk factors, African American and hispanic population (earlier age of onset of stroke and higher prevalence of atherosclerosis)? Please discuss under implications of this study.

Answer: 

We agree that based on our research, the generalizability to other settings in which predictor and outcome prevalence differ, such as in another geographic region, cannot be confirmed. Therefore, in the strengths and limitations section we state: ‘Formal external validation of the final prediction model should be conducted to confirm predictive performance over time in the Dutch OHS-PC setting or to prove its predictive performance in other settings.’

Changes to the manuscript:

To the Clinical implications this sentence was added: ‘External validation studies (temporal as well as geographical) and implementation studies are needed to evaluate the generalizability and clinical impact of these findings.’

---

## [Decision Letter · Decision Letter 1]

6 Sep 2024

Additional predictors of stroke and transient ischaemic attack in BEFAST positive patients in out-of-hours emergency primary care

PONE-D-23-43803R1

Dear Dr. van Royen,

We’re pleased to inform you that your manuscript has been judged scientifically suitable for publication and will be formally accepted for publication once it meets all outstanding technical requirements.

Kind regards,

Shashank Shekhar, MD

Academic Editor

PLOS ONE

Additional Editor Comments (optional):

Reviewers' comments:

Reviewer's Responses to Questions

**Comments to the Author**

1. If the authors have adequately addressed your comments raised in a previous round of review and you feel that this manuscript is now acceptable for publication, you may indicate that here to bypass the “Comments to the Author” section, enter your conflict of interest statement in the “Confidential to Editor” section, and submit your "Accept" recommendation.

Reviewer #1: All comments have been addressed

Reviewer #2: All comments have been addressed

2. Is the manuscript technically sound, and do the data support the conclusions?

Reviewer #1: Yes

Reviewer #2: Yes

3. Has the statistical analysis been performed appropriately and rigorously? 

Reviewer #1: Yes

Reviewer #2: Yes

4. Have the authors made all data underlying the findings in their manuscript fully available?

Reviewer #1: Yes

Reviewer #2: Yes

5. Is the manuscript presented in an intelligible fashion and written in standard English?

Reviewer #1: Yes

Reviewer #2: Yes

6. Review Comments to the Author

Reviewer #1: Very useful tool for clinicians to stratify patients at risk for stroke. It shall be broadly. disseminated

Reviewer #2: (No Response)

7. PLOS authors have the option to publish the peer review history of their article (what does this mean?). If published, this will include your full peer review and any attached files.

Reviewer #1: **Yes: **Aurel Popa-Wagner

Reviewer #2: No

---

## [Editor Report · Acceptance letter]

11 Sep 2024

PONE-D-23-43803R1 

PLOS ONE

Dear Dr. van Royen, 

I'm pleased to inform you that your manuscript has been deemed suitable for publication in PLOS ONE. Congratulations! Your manuscript is now being handed over to our production team.

Kind regards, 

on behalf of

Dr. Shashank Shekhar 

Academic Editor

PLOS ONE